# OpenReview forum: "Safety-Aware Fine-Tuning of Large Language Models"
_NeurIPS.cc/2024/Workshop/SafeGenAi — SafeGenAi Poster_

### Official Review · Reviewer_RJcR · 2024-10-09

**Rating:** 6
**Confidence:** 3

**Review:**

**Summary:** The paper presents a safety-aware fine-tuning method to detect and remove the potentially harmful data by using a scoring function that exploits the subspace information of harmful and benign samples. The harmfulness is significantly reduced by the method in different LLM.

**Strengths:**

1. Experimental results show that the proposed method significantly reduces the harmfulness in two LLM using different qualities of datasets.
2. The paper provides a detailed analysis and ablation studies.

**Weaknesses:**
1. The paper does not clearly discuss the disadvantages of manually filtering. The reviewer has doubts about the motivation of the method.
2. There is no figure that can help readers understand the method.
3. In Section 4, only the methods were outlined, lacking a description of the design motivation in the methods.

---

### Official Review · Reviewer_rPs9 · 2024-10-10
**Safety-Aware Fine-Tuning of Large Language Models**

**Rating:** 5
**Confidence:** 4

**Review:**

Summary:

The paper proposes a safety-aware fine-tuning framework for training LLMs to generate less harmful responses. The key idea is to design a filter that leverages sub-space information from LLM embeddings, to separate harmful samples in various degrees of contamination rate from the training dataset, and then evaluate the performance of the model trained on this augmented dataset. Experiments are conducted on the Beavertails classification dataset, where models trained using the proposed framework tend to generate less harmful results as per the harmfulness score.

Strengths:

-	The presented results demonstrate that models trained using the proposed framework indeed generate lower harmfulness scores as compared to traditional SFT paradigm. Moreover, this is not at a great cost to other metrics like helpfulness score.
-	The paper is generally written well, and therefore is easy to follow.

Weaknesses:

-	The intuition behind why harmful samples will be located away from the origin as opposed to benign ones being closer as shown in Figure 2b is unclear. Is the interpretation that LLMs embed harmful vs benign samples differently? If so, there needs to be a better way to interpret it than the 1 vs k dimensional example provided. Most likely, what might have happened is that since harmful samples are less common in the experimented dataset, their embeddings might appear as an outlier in the distribution, which creates this trend. In this case, this approach will not generalize to a dataset which pre-dominantly has harmful samples.

-	A deeper analysis of Table 2 results is missing. For instance, why does the SAFT sampling strategy yield better HS score than completely training on benign data? Was the size of the datasets used in SAFT vs Oracle the same? Could it be that with \lambda = 0.3, SAFT was eliminating 30% of the samples, and hence the resultant filtered dataset was much smaller than all benign samples used in Oracle? These explanations are missing in the text.

-	While the motivation behind the experiments is good, it is unclear how this approach to scale to general tasks outside the domain of harmful dataset classification. For instance, can this approach be used to train models on general web-scale datasets where annotations about harmfulness may not be as apparent as in the Beavertails dataset? While this is not a major drawback, some analysis about this in the paper (besides the mention in the Broader Impact in Appendix) would be a good addition.

---

### Official Review · Reviewer_tA5j · 2024-10-10

**Rating:** 5
**Confidence:** 4

**Review:**

**Summary**

This paper proposes a method SAFT, which filters out harmful data when doing SFT on LLMs. SAFT is built upon the assumption that harmful data can lead to a change in activation in a consistant manner, compared to benigh data. With matrix factorization, SAFT can discover harmful data and remove them from fine-tuning, leading to less harmful and more helpful LLMs compared to standard SFT.


**Strength**
- The paper tries to filter out harmfull data during fine-tuning of LLMs, which is a very important problem.

- The paper proposes an efficient and practical solution, supported by results improved harmlessness on LLAMA2 and Vicuna.

- The paper provides lots of analysis, such as contamination ratio, helping users better understand the approach.

**Weakness**
- The definitions of helpfulness is limited. Specifically, helpfulness is measured by BLEURT and ROUGE-L, metrics that computes lexical overlap. While helpfulness is beyond lexical overlap.

- I assume the harmfulness of data samples is highly non-linear, given the diversity of language and semantics. However, the proposed approach is a matrix decomposision-based approach. I'm a bit concerned that this restricts the effectiveness of discovering harmful data and the applicability of the method to a large extent.

- It is not very clear whether embedding $z$ refers to the shallow embedding or the contexualized embedding? If the latter one, it is better to refer to it as the hidden states from the last layer. What if you use Transformer encoder-only model, or encoder-decoder model?